# Observational evidence for on-shelf heat transport driven by dense water export in the Weddell Sea

Elin Darelius [1] ✉, Kjersti Daae [1,6], Vår Dundas[1,6], Ilker Fer [1,6], Hartmut H. Hellmer [2,6], Markus Janout [2,6], Keith W. Nicholls [3,6], Jean-Baptiste Sallée[4,6] & Svein Østerhus [5,6]

The transport of oceanic heat towards the Antarctic continental margin is central to the mass balance of the Antarctic Ice Sheet. Recent modeling efforts challenge our view on where and how the on-shelf heat flux occurs, suggesting that it is largest where dense shelf waters cascade down the continental slope. Here we provide observational evidence supporting this claim. Using records from moored instruments, we link the downslope flow of dense water from the Filchner overflow to upslope and on-shelf flow of warm water.

The ice shelves fringing the Antarctic continent are thinning at an accelerating rate[1], causing the feeding ice streams to accelerate and thin. Consequently, the Antarctic ice sheet has lost >2 billion tons of ice, contributing to about 8 mm of sea-level rise since 1992[2]. For most ice shelves, basal melt, driven by ocean currents entering the ice shelf cavities, is a major sink term in their mass balance[3]. The oceanic heat originates from the Circumpolar Deep Water (CDW) that circumnavigates the continent in the Antarctic Circumpolar Current (ACC). The warm waters approach the continental shelf of the marginal seas either due to the proximity of the ACC, e.g. in the Amundsen Sea, or as part of subpolar gyres, e.g. the Weddell and Ross gyres. In the Weddell Sea, which is the focus here, the CDW mixes with relatively fresh waters and cools to form Warm Deep Water (WDW). WDW is transported along the continental slope of the southern Weddell Sea in the Antarctic Slope Current, and it is separated from the continental shelf by the Antarctic Slope Front[4].

The amount of CDW found on the continental shelf varies substantially around Antarctica. While the warm shelves in the Amundsen and Bellingshausen Seas are flooded with CDW, its presence is limited on the cold shelves in the Ross and the Weddell Seas, where water mass transformation and dense water formation are prominent during winter (see e.g. ref. [5]). Somewhat counterintuitive, a recent modeling study[6] suggests that while the heat content is largest on warm shelves, the on-shelf heat flux is greater in regions where dense waters are produced and exported. The heat flux is particularly localized and concentrated in areas where dense water is channeled down the continental slope by submarine ridges or canyons. On-shelf heat fluxes are typically associated with troughs on the warm shelves, e.g. refs. [7,8], and eddy-transports, e.g. refs. [9,10]. As stated by the authors, the results presented by Morrison et al.[6] signify a shift in our understanding of where and how CDW approaches and enters the Antarctic continental shelves. Large on-shelf heat fluxes in cold shelf regions, such as the Weddell Sea, underline the importance of water mass modification and atmospheric interaction on these shelves[11] in shielding cold region ice shelves from oceanic heat. While there are examples of CDW intrusions being co-located with dense shelf water export, e.g. refs [12,13], the process for on-shelf transport suggested by Morrison et al.[6] is hitherto unobserved.

The modeling results suggest that the topographically steered, downslope flow of dense water is associated with a depression in the sea surface height (SSH) that is aligned with the corrugation supporting the downslope flow (Supplementary Fig. 1). The negative SSH anomaly, which in their simulations has a horizontal scale of 20 km, drives a barotropic (depth-independent) cross-slope flow that enhances the downslope flow of dense water west of the SSH anomaly. On the eastern side of the anomaly, an oppositely directed—i.e. *upslope*—current is set up. The upslope current transports CDW shoreward and is responsible for the localized on-shelf heat flux in the numerical model simulations. A similar mechanism was proposed by Kämpf[14].

[1]Geophysical Institute, University of Bergen and the Bjerknes Centre for climate Research, Alleg. 70, Bergen, Norway. [2]Alfred Wegener Institute, Helmholtz Centre for Polar and Marine Research, Am Handelshafen 12, Bremerhaven, Germany. [3]British Antarctic Survey, High Cross, Madingley Road, Cambridge, UK. [4]Sorbonne Université, CNRS, LOCEAN, 4, Place Jussieu, Paris, France. [5]NORCE Norwegian Research Centre AS and the Bjerknes Centre for Climate Research, Jahnebakken 5, Bergen, Norway. [6]These authors contributed equally: Kjersti Daae, Vår Dundas, Ilker Fer, Hartmut H. Hellmer, Markus Janout, Keith W. Nicholls, Jean-Baptiste Sallée, Svein Østerhus. ✉e-mail: elin.darelius@uib.no

In the southern Weddell Sea, such a topographically steered, downslope flow of dense water has been observed [Supplementary Fig. 1 and refs. [15,16]] in the vicinity of a ridge that cross-cuts the continental slope east of the Filchner Trough (FT, Fig. 1) opening. The dense water consists of Ice Shelf Water (ISW, $T < T_F$, where $T_F$ is the surface freezing point), formed within the Filchner-Ronne Ice Shelf cavity. ISW spills over the FT sill at a rate of 1.6 Sv (1 Sv = $10^6 m^3 s^{-1}$) and flows westward as a dense bottom plume in the upper continental slope[16]. Along its path, where the plume impinges on the first of the two submarine ridges that cross-cut the slope about 80 km west of the FT opening (Fig. 1), a fraction of the plume is steered downslope by the ridge. Observations from an instrumented mooring at a position in the vicinity of the ridge (D1 at 2100 m depth, Fig. 1d) show a vigorous (speeds up to 1 m s$^{-1}$) downslope flow of ISW parallel to the ridge[15,16]. In addition, a cross-ridge CTD-section shows a layer of ISW, seemingly leaning against the ridge [See Supplementary Fig. 1 and ref. [15]].

In the following, we provide observational evidence supporting the claims by Morrison et al.[6], i.e., that the topographically steered, downslope flow of dense water along the ridge is associated with an upslope flow further to the east, that brings Warm Deep Water (WDW), the Weddell Sea version of CDW, up the slope where it enters the continental shelf.

## Results

### Mean currents

East of the Filchner Trough, moorings deployed on the upper part of the slope capture the westward-flowing Antarctic Slope Current[17]. Downstream of the trough, moorings have typically been instrumented to study the dense plume, but most of them extended into the overlying WDW. The mean flow within the plume (blue arrows in Fig. 1d) is roughly aligned with isobaths—as expected from a dense gravity current under the influence of rotation[18,19]. The exception is D1, where the mean current is steered downslope by the topography[15,16]. At moorings upstream of the ridge, the mean currents observed above the plume (red arrows in Fig. 1d, see methods), in general, show that the flow has an upslope component. The largest upslope current (0.05 m s$^{-1}$) is observed at a mooring located roughly 25 km east of the ridge at 1800 m depth (D2, Fig. 1d), in close agreement with the 20 km offset found in the simulations by Morrison et al.[6]. Moorings closer to the ridge show weaker (0.01–0.02 m s$^{-1}$) upslope currents within the WDW. The upslope flow of WDW, according to Morrison et al.[6], would be barotropic and expected to extend to the plume layer below. The observed baroclinic, alongslope (i.e., parallel to isobaths) component is however larger and dominates the flow within the plume.

### Correlation between down- and upslope flow

The numerical simulations[6] show a high temporal correlation between the downslope transport of dense water and the upslope transport of warm water. While the mooring records do not allow for a transport estimate, we can analyze the correlation between the cross-slope component of the velocity recorded by the moorings. The only two concurrent mooring records (D1 next to the ridge and D2 25 km to the east, see Fig. 1d) are dominated by continental shelf waves of different periods, 6 days at D1 vs. 3 days at D2[17,20]. Still, they show significant coherence on tidal frequencies and, more relevant here, on time scales longer than about 15 days (Fig. 2a). The 10-day low-passed filtered

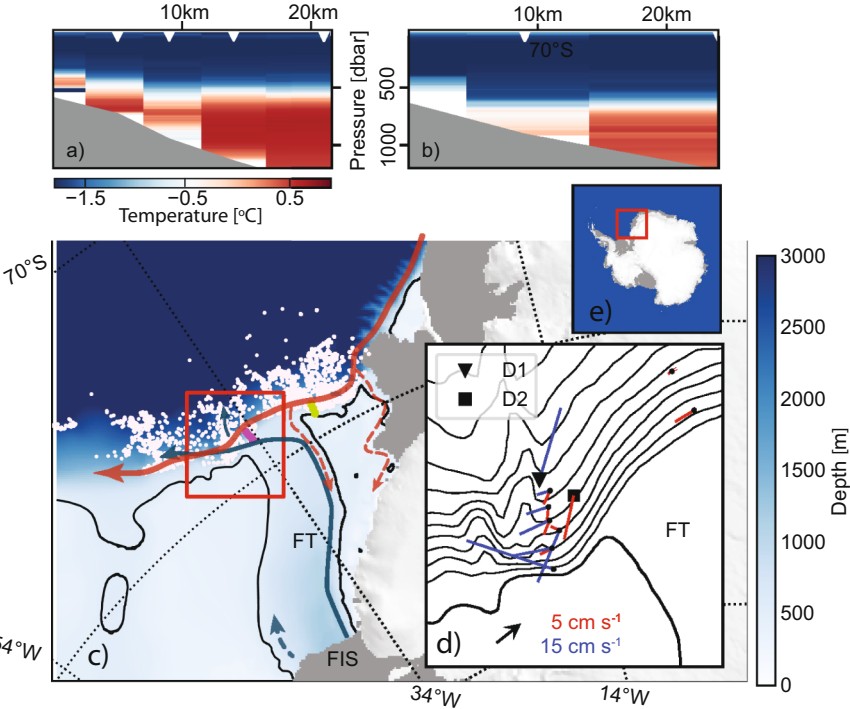

**Fig. 1 | Map over the study area and temperature sections.** Temperature sections **a** west and **b** east of the Filchner Trough (purple/green line in **c**, respectively). The divergent colormap highlights the depth of the thermocline, with Winter Water above (in blue) and (modified) Warm Deep Water below (in red). **c** Map over the study area with bathymetry[27] in blue shading according to the color bar and the 500 m isobath highlighted in black. The circulation in the area is shown with blue (outflow of Ice Shelf Water) and red (the Antarctic Slope Current and the Antarctic Coastal Current) arrows[34,35]. Dashed arrows indicate seasonal flow[35,36]. Floating ice shelves are shown in dark gray and land in light gray. The location of Conductivity-Temperature-Depth (CTD) stations on the continental slope is marked with white dots. FT indicates the Filchner Trough, and FIS is the Filchner Ice Shelf. **d** Mean current from moorings listed in Table 1. Currents observed within the dense plume are shown in blue, and currents observed above/outside the plume are shown in red. The black arrow on the shelf gives the velocity scale. Note that there is a factor three difference between the blue and red scales. Isobaths are shown every 250 m with the 500 m isobath in bold, and the area is marked with a red rectangle in **d**. **e** A map showing the location of the study area in the southwestern Weddell Sea (red square).

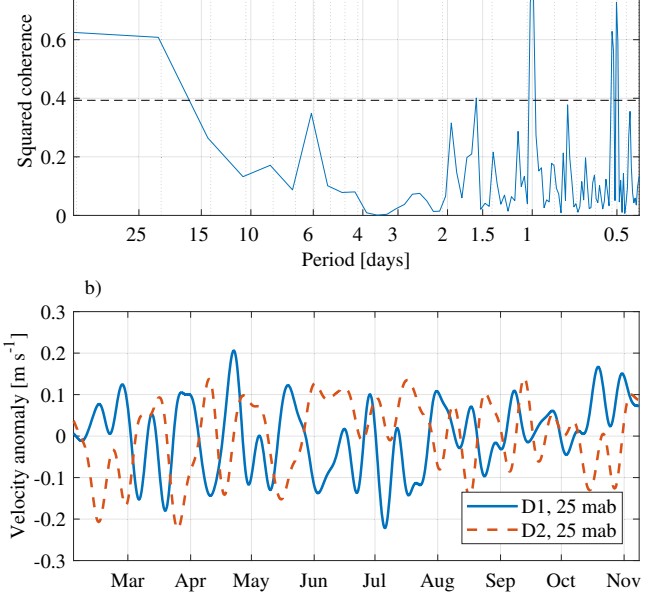

**Fig. 2 | Observed cross-isobath velocities. a** Squared coherence between the D1 and D2 (25 meters above bottom, mab) current component aligned with the mean current at D1, see Fig. 1d. The dashed black line shows the 95% confidence level. **b** 10-day low-passed time series of velocity anomalies (same component as in **a**) from D1 and D2 (25 mab).

records show significant ($p < 0.01$) anti-correlation ($r = -0.65$), with the highest correlation found for a lag of 3 days with D2 leading (See Methods, Fig. 2b and Supplementary Fig. 2).

### Thermocline shoaling

Two sections of temperature (Fig. 1a, b), obtained across the upper part of the slope during a cruise in 2017, show that the WDW core is shallower in the water column and extends further onto the continental shelf west of the FT, in the vicinity of the ridge, than east of the trough. While these observations suggest an upslope motion of the WDW, the high variability in thermocline depth induced by the continental shelf waves in the area[17] prohibits conclusions based on single sections or stations. When synthesizing all available CTD data from the region, however, the pattern is confirmed: moving westward along the slope, the thermocline rises relatively sharply at about 34°W, and it is on average 200 m shallower west of the FT, in the plume region next to the ridge than further east (Fig. 3a–c). Finally, temperature profiles collected in 2011[21] by instrumented Weddell seals just south of the continental shelf break, show temperatures up to 1 °C warmer west of the trough, and the warm water generally extends higher in the water column (Fig. 3d, e).

### Discussion

The observations presented above support the mechanism proposed by Morrison et al.[6]: the downslope current along the ridge (D1) is accompanied by an upslope current to the east (D2 and others), and the strength of these currents anti-correlates, i.e. downslope flow along the ridge is associated with upslope flow further to the east. We hypothesize that the observed 2–3 day lag (Supplementary Fig. 2) is linked to the fact that we are correlating velocities from point measurements at different isobaths while Morrison et al.[6] compares up and downslope transports.

As a result of the upslope current, the WDW in this region shoals, and the thermocline is located higher in the water column near the ridge than east of the FT. According to the thermal wind equations, the westwards shoaling of the thermocline would result in a downslope, baroclinic current of 0.01–0.02 m s⁻¹ (see methods). The observed,

coherent upslope flow of WDW is consistent with an SSH anomaly and a barotropic cross-isobath flow like that suggested by Morrison et al.[6], and the barotropic component must hence be larger than the baroclinic component induced by the alongslope gradients.

An alternative explanation for the observed westward shoaling of the thermocline is that it would be linked to the interaction between the Antarctic Slope Current[4] and the opening of the FT. Idealized modeling with a setup representing the area, albeit with a smooth continental slope[22] (see methods for a brief description of the model) suggests, however, that the interaction with the FT by the current causes an increase rather than a decrease in thermocline depth, and that the effect is an order of magnitude smaller than in the results presented here (Fig. 4). This is consistent with Allen et al.[23], who show that the interaction of a current flowing in the direction of Kelvin wave propagation (i.e., westward around Antarctica) with a canyon leads to downwelling.

Mesoscale activity can also create a meridional heat flux across the slope [10]. Indeed, the temperature and velocity records at the moorings often co-vary (see e.g. ref. [20]), suggesting that there is a net horizontal, and potentially upslope, heat flux associated with the mesoscale variability. The observed temperature flux $\mathbf{Q} = (\overline{u'T'}, \overline{v'T'})$, where $x' = x - x_{Lowpassed}$ and the overline denotes time mean over the deployment length, typically has an upslope component that, within the plume region, is on the order of 0.01 K m⁻¹. This is comparable to observed values from e.g. the Denmark Strait overflow [24]. Away from, or well above, the plume, the observed temperature flux is at least one or two orders of magnitude smaller. While we are unable to separate the rotational temperature flux, which is dynamically unimportant, from the divergent temperature flux (see e.g. ref. [25]) in the observations, the results suggest that, as elsewhere, shelf waves and eddies contribute to the down-slope transport of dense, cold, plume water. We cannot exclude the possibility that eddy heat fluxes also contribute to the observed shoaling of the thermocline west of the Filchner Trough, but it is beyond the scope of this study to determine their relative contribution compared to the observed mean, southward flow in the WDW layer east of the ridge (Fig. 1d) associated with the process suggested by Morrison et al[6]. We note that in the idealized modeling study by Stewart et al.[10], which qualitatively represents the Weddell Sea (west of the FT), *all* the southwards transport of warm water is eddy-driven. However, this model configuration does not include corrugations on the continental slope that can support mean across-isobath flow outside the boundary layers. On the other hand, the idealized model used by Daae et al.[22] resolves eddy dynamics and includes both the FT and the dense outflow (but not the ridges). An analysis of the model fields from Daae et al.[22] shows no evidence of a thermocline shoaling associated with the dense outflow, despite the presence of eddies in the simulations (Fig. 4). The absence of a westward thermocline shoaling indicates that the role of eddies in driving a southward flow is relatively minor and that, in accordance with Morrison et al.[6], the ridge and the downslope flow along it are central for the upslope transport of warm water.

In conclusion, the observations suggest that the mechanism proposed by Morrison et al.[6] is at play in the Filchner overflow region and that the downslope flow of dense shelf water is driving an up-slope and on-shelf transport of WDW. The direct coupling between the outflow of dense water and the on-shelf flow of WDW may give rise to unexpected feedbacks, as, in this region, WDW serves as the source water for the formation of the dense shelf waters[26]. The transport capacity of the first FT ridge, i.e., the maximum amount of topographically steered, downslope flow of dense water the ridge can support, was estimated by Darelius and Wåhlin[15] to be 0.3 Sv. Assuming that the associated on-slope flow of WDW is of the same order, as suggested by Fig. 3 in Morrison et al.[6], it amounts to about 4% of the total circumpolar CDW flux in the simulations by Morrison et al.[6], and all (see methods) of the WDW flux required to feed the local dense water production[26].

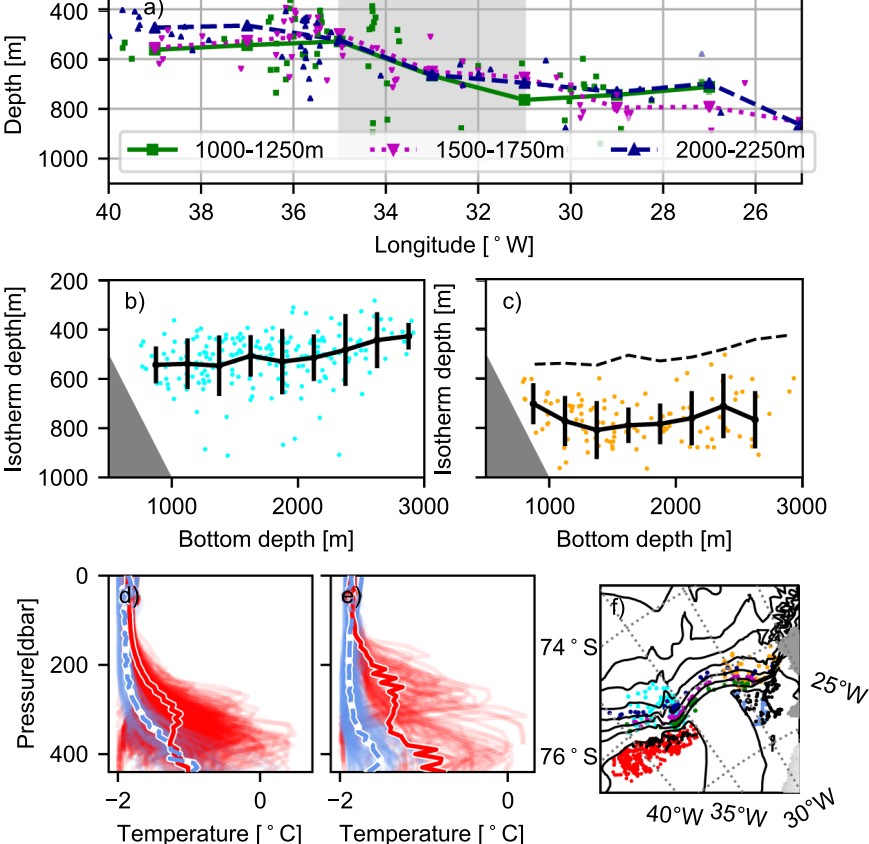

**Fig. 3 | Thermocline depth above the slope and on the shelf. a** Depth of the 0.5 °C isotherm from all available temperature profiles as a function of longitude in bottom depth ranges according to the legend. The small markers indicate an individual profile, and the lines show the mean value in 2-degree longitude intervals. The position of the profiles is indicated (in the corresponding color) in **f**. The gray shading indicates the longitudinal range of the Filchner Trough (FT). Depth of the 0.5 °C isotherm as a function of bottom depth **b** west and **c** east of the FT. The black lines in **b** and **c** show the mean isotherm depth and the standard deviation (vertical bars) in 250 m depth bins. The mean depth from **b** is included in **c** as a dashed line for reference. Temperature profiles from **d** March and **e** May collected by Weddell seals in 2011[21] on the continental shelf west (red) and east (blue, dashed) of the FT. Individual profiles are shown in pale colors, and the mean profiles are highlighted. **f** Map showing the position of the profiles in **a–d** with dots in the corresponding color and profiles from **e** as black dots. Isobaths [every 500 m][27] are shown in black.

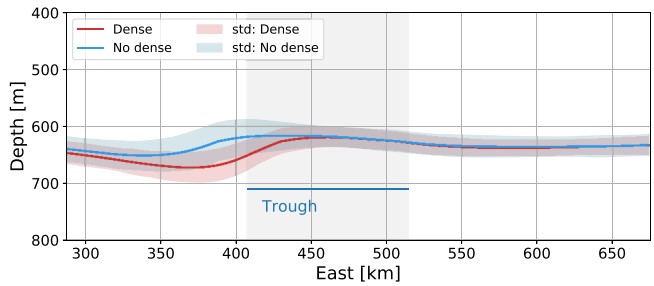

**Fig. 4 | Modeled thermocline depth.** Mean thermocline depth along the 1000-m isobath for winter runs using the idealized model described by Daae et al.[22] with (red line, Dense) and without (blue line, No Dense) dense water outflow from the idealized Filchner Trough. The position of the trough is marked in grey shadings, and the envelopes give the standard deviation (std).

## Methods

### Conductivity-Temperature-Depth (CTD) profiles

Historical CTD-profiles from the slope region north of the Filchner Trough (roughly 72–74°S, 25–45°W) were obtained from Pangaea, BODC, Coriolis, Seanoe, World Ocean Database, and WOCE; see list below for references to the data sets included. CTD profiles collected by instrumented Weddell seals during 2011[21] over the continental shelf east and west of FT were downloaded from www.meop.net. The

bottom depths at the profile locations were interpolated from the BEDMAP2 dataset[27]. Where the maximum depth of the CTD profile exceeds the bottom depth from BEDMAP2, it is used as the bottom depth for that station. The dataset from the slope comprises over 2000 profiles between 1973 and 2021, of which instrumented seals collected the vast majority (1700 profiles). However, very few of the seal profiles from the slope are deep enough to reach the temperature maximum, and hence most of the profiles used in Fig. 3a–c were obtained by ships during the austral summer (January–February).

Temperature and salinity are generally reported in Conservative Temperature, Θ, and Absolute Salinity $S_A$[28], but for the profiles collected by seals (Fig. 3d, e), in situ temperatures are shown, because many of the profiles lack salinity, which is necessary to calculate Θ.

### Mooring records

The moorings included in the study are moorings D1-2, F1-4, W2-3 and M1-2 described by Semper and Darelius, their Table 1[29]. To calculate the mean current within the plume, we use data from the deepest level with current observations. Only records where the nearest temperature record is below −0.5 °C more than 80% of the time (marked with an asterisk in Table 1) are included in Fig. 1. The mean current above the plume is approximated using the observations from the shallowest level with current measurements. Only records where the nearest temperature record is above −0.5 °C >80% of the time are included.

## Table 1 | Mooring information

| | Lon | Lat | Within plume (blue arrows) | | Above plume (red arrows) | |
|---|---|---|---|---|---|---|
| | | | T(V) height [m] | T < −0.5 °C | T(V) height [m] | T > −0.5 °C |
| F1 | 36.6°W | −74.52°S | 9 (10) | 90%* | 207 (207) | 14% |
| F2 | 36.37°W | −74.42°S | 9 (10) | 100%* | 433 (433) | 85%* |
| F3 | 36.07°W | −74.28°S | 56 (56) | 90%* | 413 (413) | 99%* |
| F4 | 35.7°W | −74.15°S | 10 (10) | 81%* | 207 (207) | 100%* |
| W2 | 36.02°W | −74.36°S | 25 (25) | 98%* | 289 (389) | 99%* |
| W3 | 35.92°W | −74.22°S | 25 (25) | 87%* | 216 (272) | 100%* |
| M1 | 32.32°W | −74.23°S | 24 (24) | 5% | 46 (46) | 96%* |
| M2 | 32.28°W | −73.98°S | 19 (19) | 0% | 156 (150) | 100%* |
| D1 | 35.75°W | −74.07°S | 25 (25) | 99%* | 100 (100) | 11% |
| D2 | 35.37°W | −74.25°S | 25 (25) | 60% | 100 (100) | 80%* |

Information about the moorings included in Fig. 1. The colors in brackets indicate the color used to plot the velocity vectors, and T/V height is the height above the bottom (in meters) of the temperature (velocity) records. Only levels marked with an asterisk (*) are included in Fig. 1.

In situ temperatures are used for the mooring records, as accompanying salinity records needed to determine Θ are often lacking.

### Thermocline depth

We approximate the thermocline depth by the depth of the (shallowest) 0.5 °C isotherm while excluding the upper 100 m of the water column with a potential solar-heated surface layer. This temperature is a few tenths of a degree lower than the WDW core temperature in this region[30], and hence representative of the depth of the relatively sharp thermocline above the WDW core. For Fig. 3b, c, the mean and standard deviation of the identified thermocline depth are calculated in bottom depth bins of 250 m width.

### Lowpass filtering and correlation

The low-pass filtered time series are filtered using a 5th-order Butterworth filter with a cut-off period of 10 days. The component shown is aligned with the mean current at D1 so that positive (negative) values indicate downslope (upslope) flow. To verify the significance of the correlation of the low-pass filtered time series, we calculated the integral timescale, $T^*$, following Emery and Thomson, their eq. 5.16[31] giving $T^* \simeq 200$ h. For a time series of length $N = 6700$ h, this gives about $N/T \simeq 70$ degrees of freedom and a critical value of 0.30 for Pearson correlations at a 99% significance level. The significance was also tested following Sciremammano[32], and that analysis suggests a critical value of 0.43. To investigate the lag, the correlation analysis was repeated using eight 60-day long, 50% overlapping windows, and the lag giving maximum correlation was identified in each window (Supplementary Fig. 2). The coherence was calculated following Emery and Thomson[31] using a window length of 1024.

### Thermal wind estimates

The alongslope gradient in thermocline depth, i.e., the westward shoaling of the WW-WDW interface along the slope, results in a vertical shear in the across-slope velocity,

$$\Delta v = -\frac{g}{\rho_0 f} \Delta \rho \frac{\Delta z}{\Delta x}. \tag{1}$$

For $\rho_0 = 1027$ kg m$^{-3}$, $\Delta \rho = 0.1$ kg m$^{-3}$, $f = -10^{-4}$ s$^{-1}$, $g = 9.82$ m s$^{-2}$, $\Delta z = 200$ m and $\Delta x = 100$–200 km (from Fig. 3a) we obtain $\Delta v = 0.01$–0.02 m s$^{-1}$ (directed down the slope).

### WDW transport estimates

Nicholls et al.[26] stated that an on-shelf transport of modified WDW (mWDW, potential temperature of $\theta = -1.76$ °C, and practical salinity of $S = 34.5$) of about 3 Sv is required to sustain the dense water production on the continental shelf. If the mWDW consists of a mixture of WDW ($\theta = 0.5$ °C) and Winter Water (WW, −1.85 °C), then <5%, or 0.15 Sv, of the mWDW would be WDW.

### Idealized model

To investigate the potential effect of (i) the interaction of the Antarctic Slope Current[4] with the opening of the FT and (ii) the role of eddy heat fluxes, we revisit the idealized numerical simulations by Daae et al.[22] that represent the study area. The model is eddy-resolving, and the domain features a smooth (i.e. no ridges) continental slope, a slope front and the associated slope current, and a widening continental shelf that is cross-cut by a trough, analogous to the FT. The model setup uses the terrain-following Regional Ocean Modeling System (ROMS, version 3.8), which is initialized using an observation-based climatology field[33]. The simulations are forced with a typical winter hydrography and a westward wind, which in the runs used here has a maximum of 3 m s$^{-1}$. For the runs with a dense outflow, the FT is initially filled with ISW, which crosses the sill at a rate of roughly 1 Sv. The flow is sustained through restoration at the southern boundary. Further details on the model setup are given in[22].

For the purpose of this study, we use monthly mean temperature fields from two runs, one with an outflow of ISW[22], refers to it as dense shelf water, DSW and one without. From these, we extract the mean depth of the 0.5°-isotherm, representing the thermocline, along the 1000 m isobath (Fig. 4). This corresponds to runs W3REF and W3HOM in ref. [22].

## Data availability

CTD data used in this study are available from www.pangaea.de, www.bodc.ac.uk, www.seanoe.org, www.coriolis.eu.org, www.ewoce.org, www.ncei.noaa.gov/products/world-ocean-database, and www.meop.net. CTD data from cruise ES006 (2003) are not searchable but are available from BODC on request (under accession number BAS220012).Mooring data used in this study are available from www.pangaea.de. The doi of individual observational data sets included in the study are listed below.

The model output files used in this study are available in the Norstore database under accession code https://archive.norstore.no/pages/public/datasetDetail.jsf?id=10.11582/2017.00003.

**Doi of mooring records:**
https://doi.org/10.1594/PANGAEA.792882https://doi.org/10.1594/PANGAEA.792883https://doi.org/10.1594/PANGAEA.792884https://doi.org/10.1594/PANGAEA.792885https://doi.org/10.1594/PANGAEA.869799https://doi.org/10.1594/PANGAEA.871146https://doi.org/10.1594/PANGAEA.869820https://doi.org/10.1594/PANGAEA.869799

**Doi of CTD data:**
https://doi.org/10.17882/54012https://doi.org/10.1594/PANGAEA.61240https://doi.org/10.1594/PANGAEA.527319https://doi.org/10.1594/PANGAEA.527410https://doi.org/10.1594/PANGAEA.527497https://doi.org/10.1594/PANGAEA.527593https://doi.org/10.1594/PANGAEA.527643https://doi.org/10.1594/PANGAEA.527812https://doi.org/10.1594/PANGAEA.734988https://doi.org/10.1594/PANGAEA.735189https://doi.org/10.1594/PANGAEA.735530https://doi.org/10.1594/PANGAEA.742577https://doi.org/10.1594/PANGAEA.742579https://doi.org/10.1594/PANGAEA.742581https://doi.org/10.1594/PANGAEA.833299https://doi.org/10.1594/PANGAEA.854148https://doi.org/10.1594/PANGAEA.859040https://doi.org/10.1594/PANGAEA.897280https://doi.org/10.1594/PANGAEA.527645-527691*

* Datasets from Pangaea in the range 527645–527691 are included in the study.

## Code availability

The Python and Matlab code used to produce the figures in the paper can be obtained from the authors on request.

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

## Acknowledgements

This study have received funding from the Norwegian Research council work through project 267660 (E.D., K.D.), from the European Union's Horizon 2020 research and innovation program under grant agreement N°821001 (J.-B.S., S.Ø.) and N°820575 (S.Ø.). Suggestions and comments by Prof. Anna Wåhlin, Prof. Marie-Noelle Houssais, and S. Semper were much appreciated. The numeric simulations were performed on resources provided by Sigma2—the National Infrastructure for High Performance Computing and Data Storage in Norway, under the project NN9608K (K.D.).

## Author contributions

E.D. analyzed the observational data and wrote the ms. K.D. set up and ran the numerical model, and V.D. analyzed the model output. E.D., I.F., H.H.H., M.J., K.N., J.-B.S., and S.Ø. have been responsible for field

activities where much of the data used in the study were collected. E.D., K.D., V.D., I.F., H.H.H., M.J., K.N., J.-B.S., and S.Ø. have discussed and revised the ms.

## Funding

## Competing interests
The authors declare no competing interests.
