## [Peer Review File · Nature Communications]

Observational evidence for on-shelf heat transport driven by dense water export in the Weddell SeaREVIEWER COMMENTS

Reviewer #1 (Remarks to the Author):

The following comments are also included in the attached pdf.

The manuscript by Darelus et al. 2022 presents observational evidence of the collocation of on-shelf (upslope) Warm Deep Water (WDW) transport and off-shelf (downslope) dense water export. These results are based on mooring, CTD-sections and historical CTD (primarily using MEOP) data collected near the Filchner Trough in the Weddell Sea.

As the authors state, this provides the first observational evidence in support of recent modeling results (from Morrison et al. 2020) of a topographically-confined on-shelf and off-shelf transport. This provides much needed insight on the pathways of dense water export and heat transport towards the Antarctic ice sheet.

Overall Recommendation:

The study is methodologically sound and the presentation is sufficiently well-organized. However, I have one major concern with respect to the conclusion presented and a number of minor comments. If addressed, I believe the work will be a suitable contribution to this journal.

Major Comments:

My major concern is as follows. With regards to the coincident inflow-outflow pairing of WDW and ISW conclusion, I am concerned about the possibility that the zonal thermocline gradient (between the two sections in Fig. 1(a) and (b)) is being partially set up by a combination of both mean and eddying components of cross- and along-isobath flow. Is the zonal thermocline gradient (being 200 m shallower west of the FT) consistent with 0.3 Sv maximum downslope transport (as mentioned in line 157) or a different transport? Might it be informative to present and discuss whether the zonal thermocline gradient is consistent with these previous transport estimates? What might the role of eddies be in both the cross- and along-isobath transport as well as the zonal thermocline gradient (perhaps the model can be used to shed more light on this)? In part, I ask this because there is not much discussion of the partition between cross- and along-isobath flow north of the trough while Fig. 1e seems to suggest that the two components might be of similar magnitudes/comparably important just downstream of the trough. Perhaps you can discuss this in the context of the model results similarly to how you present and refute in Line 135 the possibility of the ASF Current and FT interacting with this thermocline depth.

Line 117-118: Can you comment more here or in the SI why it is not possible to filter out the shelf wave frequency band in the mooring data to observe the downslope/upslope temporal correlation. Also, do the complementary numerical simulations not observe this wave spectra because of resolution? To be clear, are you saying here that the mooring data does not directly support the main conclusion of the paper and only by synthesizing all available CTD-data (mostly MEOP data) can you confirm the results from Morrison et al. 2020? If so, this point could be made more clear in the discussion in lines 117-133.

Fig. 2: A few things could be made more easily interpretable in this figure. In panel (a), it was not immediately clear what the shaded region represents (is it the longitudinal range of the trough? If that's the case, please add a label). In addition, the caption says these CTD profiles were from between the 1250 and 1500 m isobath. I interpreted this as referencing the white dots (instead of "black dots") in Fig 1d (be specific about this panel) as there are only two black dots (mooring) in panel 1e. Perhaps you could highlight the dots used here in Fig. 1(d) by adding the 1250 and 1500 m isobaths and

highlighting the region or altering the dot color within this band. This would also help with interpretability of Fig. 2(a).

It was a bit unclear why you chose a much smaller isobath range for panel (a) compared to the more inclusive range for panels (b) and (c). Could (a) for instance have two/three isobath ranges (with scatterplots and trendlines in two/three different colors)?

The data in (d) and (e) could be presented in a more visually effective manner – it currently appears as blobs. Perhaps something like mean/STD or median/IQR of the profiles for each region?

See below for minor comments on Fig. 2.

Minor Comments:

Line 122: misspelling of vicinity

Line 154: "Nicholls et al (2009)" should be in parentheses

Fig. 1: Caption line 212: Change to "The location of CTD-stations is marked with white points". The colorbar for panels (b) and (c) is missing Theta (deg. C).

Line 218: Use "because" instead of "since"

Fig. 2 (a few picky points): Panel (a) is missing units label of deg. W for Longitude. In panels (b) and (c), "Bottomdepth" should not be one word.

Panel (f) should have axes ranges and labels. If possible, please reduce the size of the red and blue dots in panel (f) as they currently appear as patches instead of distinct dots. Please also label/state in the caption the isobath spacing (I believe it has greater spacing than Fig. 1). It was not immediately clear that you had also plotted the dots from panel (a) in (f) as well as this was not in the captions. Moreover the three hues of magenta, violet, and pink are not doing these panels any service (it is particularly difficult to distinguish the pink and magenta dots in panel (f) for instance). Insert period at end of Fig. 2 caption.

Line 301: Is "doi:s" a typo?

Line 302: "We" should not be capitalized

Line 306-307: "Only records where the nearest..." is repeated from lines 303

Reviewer #2 (Remarks to the Author):

Review of Observational evidence for on-shelf transport of Warm Deep Water driven by dense water export in the Weddell Sea by Darelus et al.

This study uses observations from eastern Weddell Sea to investigate onshore (i.e. across the Antarctic shelf break) transport of Warm Deep Water (WDW), the regional version of Circumpolar Deep Water (CDW). The work is motivated by a recent modelling study that finds a close regional link between Dense Shelf Water (DSW) export and CDW inflow. The finding from the modelling study changes profoundly our view on cross-shelf heat transport and where it occurs. The present study provides the first observational evidence for the mechanism presented in the previous model study. The conclusions are based on

(i) temperature observations which reveal a shallower upper limit of the WDW layer in the vicinity of the DSW outflow compared to the continental shelf break further east where no DSW export exists, and

(ii) the onshore direction of the velocity vector in the WDW layer with the largest

velocities just east (upstream) of the DSW outflow, matching the expectations from the modelling work.

The manuscript is concise and well written, the figures are appropriate, and the data and methodology are described in detail. While the data interpretation and conclusion are robust, my main comment is on the missing temporal correlation between the DSW outflow and WDW inflow, which I detail below, followed by specific and technical comments.

Main comment

L114-118: The missing temporal correlation between the downslope and upslope flow is surprising as it is an important argument in Morrison et al. (2020). The conclusion of the study would be more robust if the mismatch in the temporal variability could be resolved. In the model, the temporal correlation is largest on the subdaily timescale, do the moorings capture this timescale? Can you discuss why there is no temporal correlation in the observations? Did you try filtering the data to remove the dominating imprint of the continental shelf waves and then calculate the correlation? Do the continental shelf waves promote any onshore WDW transport in the Weddell Sea?

Specific comments

L057-058: On the statement that the Weddell and Ross Gyres move CDW closer to the continental shelf. CDW can also be found elsewhere on the continental shelf (e.g. Amundsen and Bellingshausen Seas, or near the Totten Ice Shelf in East Antarctica). How does this fit to the statement?

L063-064: Please add reference for dense water formation.

L075-076: (i) Do observations of other onshore CDW transport processes exist? Adding the information here might help with my comment on L159-160.

(ii) Morrison et al. (2020) discuss that Orsi et al. (2009) and Williams et al. (2010) report concurrent CDW inflow and DSW outflow. Can you comment on those observations and acknowledge them if you assess them as insightful?

L103-105: Can you comment on the fact that most of the flow captured by the CTD section is in along-slope direction (and not downslope)? Does this matter?

L108-109: The moorings show along-slope flow at depth and onshore flow in the WDW layer. Can you comment on the co-location of the flow? The mechanism described in the text so far suggests a spatial offset between downslope ISW and upslope WDW transport.

L111-112: Do you expect to find the maximum upslope velocity to be at this location? Or is it impossible to make a statement on this due to lack of observations in the area? The 25 km is close to the 20 km that Morrison et al. (2020) find.

L127: Suggest adding 34°W longitude to Fig. 1 to help with orientation.

L148-151: Please add a statement on the missing temporal correlation between the downslope and upslope flow.

L159-160: Where does the remainder (90%) of the WDW-flux occur? At the eastern part of the FT as described in Darelius et al. (2004) and Ryan et al. (2017)? Morrison et al. (2020) suggest the majority of the onshore CDW transport is connected to the DSW export, do the observations presented here support the model findings?

L180-181: Do you expect the summer bias of the CTD observations to affect the findings of this study?

L340: Is the slope front current different to the ASC? If not, please use the same name throughout the manuscript.

Figure 1

- Colormap for Fig 1a-b): Suggest explaining in figure caption that -0.5°C represents the threshold between ISW and WDW and that is why a divergent colormap was chosen which is centred around -0.5°C .
- Fig 1d): Grey temperature section in Fig. 1d) is difficult to see, would a different colour (green?) improve the visibility?
- L212: The location of CTD-stations on the continental slope is marked. Are they marked by the white dots? Please specify and mention what the white dots represent otherwise.
- Suggest adding explanation of mooring locations (white circle and square) in Fig. 1e) to figure caption.

Figure 2

- What is the shading in Fig 1a) indicating? Please describe in figure caption.
- What are the black dots in Fig. 1f)?

Figure S1

- Should the minimum SSH not be centred around the dense water plume? (Assuming conservative temperature serves as a tracer for the dense plume.)
- The section shown is oriented across the continental slope and therefore approximately orthogonal to the shelf break. The surface pressure argument brought forward in Morrison et al. (2020) and described in L077-085 assumes a SSH change across a trough/ridge (parallel to the shelf break). Why is the SSH shown schematically in Fig. S1a) relevant?

Technical comments

L069-070: Check reference format

L097: It should be Fig. 1e)

L110: Add info on panel: Fig. 1e). Suggest adding information that upslope flow is shown in red.

L112: It should be Fig. 1d)

L119: Add info panel: Fig. 1b-c)

L121: Correct to: extends

L122: Correct to: vicinity.

L132: Delete with east of the

L136: Remove Front

L152: Rewrite to: on-shelf flow

L159: Does the total flux refer to the total circumpolar flux?

L302: Remove capital letter in we

L320-321: Rewrite to: Antarctic Slope Current (capital letters) and remove ASC acronym

unless it is used in the manuscript.

Figure 1

L210: Replace slope front with slope current or Antarctic Slope Current (see comment on L340)

Please change the brackets of the axis labels to square brackets for consistency with other figures.

Figure 2

L261: Specify panel where black dots are shown: Fig. 1d)

L 264: Rewrite to: Temperature profiles from d) March and e) May

L266: Rewrite to: f) Map showing

Figure S1

Please use larger font size.

Please change the brackets of the axis labels to square brackets for consistency with other figures.

Figure S2

I suggest being consistent with the orientation of the vertical depth coordinate, i.e. change the depth values to positive as in Fig 1-2.

Table 1

L323: Consider highlighting T(V) ins. height in table caption in italic or asterisks to improve readability.

Mentioned (new) references

Orsi, A. H., Wiederwohl, C. L. (2009), A recount of Ross Sea waters. *Deep-Sea Res. II* 56, 778–795, doi: 10.1016/j.dsr2.2008.10.033.

Williams, G. D., S. Aoki, S. S. Jacobs, S. R. Rintoul, T. Tamura, and N. L. Bindoff (2010), Antarctic Bottom Water from the Adélie and George V Land coast, East Antarctica (140–149°E), *J. Geophys. Res.*, 115, C04027, doi:10.1029/2009JC005812.

Bergen, 18 November 2022

Dear reviewers,

Thank you for your thorough reviews of our work and suggestions for improving the manuscript. We have revised the manuscript accordingly, and you will find point-by-point answers (in italic) to your comments below.

Your primary concern was the lack of correlation between the upslope and downslope currents and that we did not investigate the potential role of eddies in setting up the observed thermocline slope – both these issues are, as described in more detail below, now addressed; We show a correlation at longer time scales and, based on the numerical simulations, we suggest that the role of eddies is relatively minor.

Best regards,

Elin Darelius and co-authors

REVIEWER COMMENTS

Reviewer #1 (Remarks to the Author):

The following comments are also included in the attached pdf.

The manuscript by Darelius et al. 2022 presents observational evidence of the collocation of on-shelf (upslope) Warm Deep Water (WDW) transport and off-shelf (downslope) dense water export. These results are based on mooring, CTD-sections and historical CTD (primarily using MEOP) data collected near the Filchner Trough in the Weddell Sea.

As the authors state, this provides the first observational evidence in support of recent modeling results (from Morrison et al. 2020) of a topographically-confined on-shelf and off-shelf transport. This provides much needed insight on the pathways of dense water export and heat transport towards the Antarctic ice sheet.

Overall Recommendation:

The study is methodologically sound and the presentation is sufficiently well-organized. However, I have one major concern with respect to the conclusion presented and a number of minor comments. If addressed, I believe the work will be a suitable contribution to this journal.

Major Comments:

My major concern is as follows. With regards to the coincident inflow-outflow pairing of WDW and ISW conclusion, I am concerned about the possibility that the zonal thermocline gradient (between the two sections in Fig. 1(a) and (b)) is being partially set up by a combination of both mean and eddy components of cross- and along-isobath flow.

To address the reviewer's concern, we have revised the manuscript to discuss further the potential role of eddies in setting the zonal thermocline gradient. We now acknowledge that eddy transports may contribute to the shoaling thermocline in the area, but based on the available observations, it is not possible to quantify their relative contribution. However, the results from the eddy-resolving, idealized modeling (previously Fig S2 – now Fig 4) do not show a shoaling of the thermocline associated with the dense outflow. This indicates that the role of the eddies is relatively minor.

The original ms included an observation-based estimate of the mean upslope currents (Fig 1d). In addition, we now show a correlation (New Fig. 2) between the upslope and downslope flow on timescales longer than that typical for eddies and the outflow-generated shelf waves in the regions. This implies that the mechanism proposed by Morrison et al. is at play in the area.

Suggested new text:

... the temperature and velocity records at the moorings often co-vary (see, e.g. Darelius et al, 2009), suggesting that there is a net horizontal, and potentially upslope, heat flux associated with the mesoscale variability. The observed temperature flux $Q = \overline{(u'T' + v'T')}$, where $x' = x - x_{Lowpassed}$ and the overline denotes time mean over the deployment length, typically has an upslope component that, within the plume region, is on the order of 0.01 K m s^{-1} . This is comparable to observed values from e.g. the Denmark Strait overflow (Voet and Quadfasel, 2010). Away from, or well above, the plume, the observed temperature flux is at least one or two orders of magnitude smaller. While we are unable to separate the rotational temperature flux, which is dynamically unimportant, from the divergent temperature flux (see e.g. Guo et al, 2014) in the observations, the results suggest that, as elsewhere, shelf waves and eddies contribute to the down-slope transport of dense, cold, plume water. While we cannot exclude the possibility that eddy heat fluxes contribute to the observed shoaling of the thermocline west of the Filchner Trough, it is beyond the scope of this study to determine their relative contribution compared to the observed mean, southward flow in the WDW layer east of the ridge (Fig. 1d) associated with the process suggested by Morrison et al (2020). We note that in the idealized modeling study by Stewart and Thompson (2016), which qualitatively represents the Weddell Sea (west of the FT), all the southwards transport of warm water is eddy-driven. However, this model configuration does not include corrugations on the continental slope that can support mean across-isobath flow outside the boundary layers. On the other hand, the idealized model used by Daae et al (2017) resolves eddy dynamics and includes both the FT and the dense outflow (but not the ridges). An analysis of the model fields from Daae et al (2017) shows no evidence of a thermocline shoaling associated with the dense outflow, despite the presence of eddies in the simulations (Fig. 4). The absence of a westward thermocline shoaling indicates that the role of eddies in driving a southward flow is relatively minor and that, in accordance with Morrison et al (2020), the ridge and the downslope flow along it are central for the upslope transport of warm water.

Is the zonal thermocline gradient (being 200 m shallower west of the FT) consistent with 0.3 Sv maximum downslope transport (as mentioned in line 157) or a different transport? Might it be informative to present and discuss whether the zonal thermocline gradient is consistent with these previous transport estimates?

No, the downslope transport along the ridge and the estimated transport capacity is not driven by the gradient in thermocline depth (200 m over 100 km) referred to above, but to a much sharper gradient (200 m over a few kilometers, see Supplementary Fig 1) associated with the topographically steered downslope flow of dense (and cold) plume water – see e.g.

Darelius & Wåhlin (2007) for more details. The pressure gradient associated with gradients in SSH, suggested by Morrison et al., 2020 would increase the downslope transport capacity.

The gradient in thermocline depth (i.e., the interface between the Winter water and the (m)WDW) would result in a thermal wind current $\Delta v = -\frac{g}{\rho_0 f} \Delta \rho \frac{\Delta z}{\Delta x}$

of about 1-2 cm s⁻¹ (for $\Delta \rho \sim 0.1 \text{ kg m}^{-3}$, $\Delta z \sim 200 \text{ m}$ occurring between roughly 30-36W, giving $\Delta x \sim 100 - 200 \text{ km}$, from Fig 2a) directed downslope, i.e., opposite to the mean upslope currents observed, e.g., at D2.

This is now mentioned in the ms:

According to the thermal wind equations, the westwards shoaling of the thermocline would result in a downslope, baroclinic current of 0.01-0.02 m s⁻¹ (see methods). The observed, coherent upslope flow of WDW is consistent with an SSH anomaly and a barotropic cross-isobath flow like that suggested by Morrison et al (2020), and the barotropic component must hence be larger than the baroclinic component induced by the alongslope gradients.

And the details of the calculations are given in the methods:

The along slope gradient in thermocline depth, i.e. the westward shoaling of the WW-WDW interface along the slope results in a vertical shear in the across-slope velocity,

$$\Delta v = -g/\rho_0/f^* \Delta \rho \Delta z/\Delta x . (1)$$

For $\rho_0=1027 \text{ kg m}^{-3}$, $\Delta \rho=0.1 \text{ kg m}^{-3}$, $f=-10^{-4} \text{ s}^{-1}$, $g=9.82 \text{ m s}^{-2}$, $\Delta z=200 \text{ m}$ and $\Delta x = 100 - 200 \text{ km}$ (from Fig. 3a) we obtain $\Delta v = 0.01 - 0.02 \text{ m s}^{-1}$ (directed down the slope).

What might the role of eddies be in both the cross- and along-isobath transport as well as the zonal thermocline gradient (perhaps the model can be used to shed more light on this)? In part, I ask this because there is not much discussion of the partition between cross- and along-isobath flow north of the trough while Fig. 1e

seems to suggest that the two components might be of similar magnitudes/comparably important just downstream of the trough. Perhaps you can discuss this in the context of the model results similarly to how you present and refute in Line 135 the possibility of the ASF Current and FT interacting with this thermocline depth.

We now describe/explain the observed cross- and along isobath components of the flow east of the ridge (L135 and on), and we acknowledge in the ms (L 210 and on) that eddies and shelf waves may contribute to the observed shoaling of the thermocline, but it is not possible based on the available observations - nor using the model, since it does not include the ridge - to quantify their relative contribution. We note, however, that the (eddy-resolving) model does not show a shoaling of the thermocline west of FT (Old Fig S2, new Fig 4). This indicates that the role of eddies is relatively minor.

Line 117-118: Can you comment more here or in the SI why it is not possible to filter out the shelf wave frequency band in the mooring data to observe the downslope/upslope temporal correlation. Also, do the complementary numerical simulations not observe this wave spectra because of resolution? To be clear, are you saying here that the mooring data does not directly support the main conclusion of the paper and only by synthesizing all available CTD-data (mostly MEOP data) can you confirm the results from Morrison et al. 2020? If so, this point

could be made more clear in the discussion in lines 117-133.

We were admittedly blinded by the variability linked to the shelf waves and did not think about filtering them out. Morrison et al. show a significant correlation for their 3h-resolution records of transport – but they are in a region where the bottom slope is too steep for the outflow to generate waves (Han et al., 2022) and their model does not include tides. Following your comments - and those of the second reviewer – we now present low-pass filtered currents from the two concurrent moorings (New Fig. 2) and show that there is significant coherence at periods longer than ten days and that the low-pass filtered currents are negatively correlated ($r=-0.67$, $p<0.01$ at three days lag). When there is higher than normal downslope flow close to the ridge, there is an anomalously high upslope current at the mooring 25 km east of the ridge – as suggested by the Morrison et al. paper. While it is not quantified, Fig 4c in Morrison et al. (2020) does suggest an anti-correlation of low-frequency variability in their transport records.

To sum up: The observed upslope (D2) and downslope (D1) velocities are anti-correlated and the deployment mean currents shown in Fig. 1d also agree with Morrison et al. Both mooring data and CTD data support that the mechanism suggested by Morrison et al. (2020) is at play in the Filchner region.

New text:

The numerical simulations (Morrison et al, 2020) show a high temporal correlation between the downslope transport of dense water and the upslope transport of warm water. While the mooring records do not allow for a transport estimate, we can analyze the correlation between the cross-slope component of the velocity recorded by the moorings. The only two concurrent mooring records (D1 next to the ridge and D2 25 km to the east, see Fig. 1d) are dominated by continental shelf waves of different periods (6 days at D1 vs. three days at D2 Darelius et al, 2009; Jensen et al, 2013). Still, they show significant coherence on longer time scales (>10 days). Using 10-day low-pass filtered records, we obtain a significant anti-correlation ($r=-0.67$, $p<0.01$ at three days lag), Fig. 2b.

Fig. 2: A few things could be made more easily interpretable in this figure. In panel (a), it was not immediately clear what the shaded region represents (is it the longitudinal range of the trough? If that's the case, please add a label).

The grey area indicates the longitudinal range of the FT. This is now clarified in the caption.

In addition, the caption says these CTD profiles were from between the 1250 and 1500 m isobath. I interpreted this as referencing the white dots (instead of “black dots”) in Fig 1d (be specific about this panel) as there are only two black dots (mooring) in panel 1e. Perhaps you could highlight the dots used here in Fig. 1(d) by adding the 1250 and 1500 m isobaths and highlighting the region or altering the dot color within this band. This would also help with interpretability of Fig. 2(a).

The location of the profiles included in Fig 2a-e are now marked in Fig 2f

It was a bit unclear why you chose a much smaller isobath range for panel (a) compared to the more inclusive range for panels (b) and (c). Could (a) for instance have two/three isobath ranges (with scatterplots and trendlines in two/three different colors)?

We now include data from three depth ranges, plotted in different colors as suggested

The data in (d) and (e) could be presented in a more visually effective manner – it currently appears as blobs. Perhaps something like mean/STD or median/IQR of the profiles for each region?

See below for minor comments on Fig. 2.

The figure now highlights the mean profile while showing all individual profiles in the background.

Minor Comments:

Line 122: misspelling of vicinity

Corrected

Line 154: “Nicholls et al (2009)” should be in parentheses

Corrected

Fig. 1: Caption line 212: Change to “The location of CTD-stations is marked with white points”.

Corrected

The colorbar for panels (b) and (c) is missing Theta (deg. C).

Corrected

Line 218: Use “because” instead of “since”

Corrected

Fig. 2 (a few picky points):

Panel (a) is missing units label of deg. W for Longitude.

Corrected

In panels (b) and (c), “Bottomdepth” should not be one word.

Corrected

Panel (f) should have axes ranges and labels.

Corrected

If possible, please reduce the size of the red and blue dots in panel (f) as they currently appear as patches instead of distinct dots.

Corrected

Please also label/state in the caption the isobath spacing (I believe it has greater spacing than Fig. 1).

Corrected

It was not immediately clear that you had also plotted the dots from panel (a) in (f) as well as this was not in the captions. Moreover the three hues of magenta, violet, and pink are not doing these panels any service (it is particularly difficult to distinguish the pink and magenta dots in panel (f) for instance).

We have used different colors that hopefully are more readily discernable. We now write in the caption that their position is shown in panel (f): “(f) Map showing the position of the profiles in panels (a-d) with dots in the corresponding color and profiles from panel (e) as black dots.”

Insert period at end of Fig. 2 caption.

Corrected

Line 301: Is “doi:s” a typo?

Not sure how to write doi in plural – but I guess the editorial team will pick this up and correct it if we get that far. I have now changed to “doi”.

Line 302: “We” should not be capitalized

Corrected

Line 306-307: “Only records where the nearest...” is repeated from lines 303

Yes, but the first time we describe how the mean current within the plume (blue arrows in Fig 1) is calculated, and the second time how the current above the plume (red arrows in Fig 2) is calculated, so the wording is slightly different.

Reviewer #2 (Remarks to the Author):

Review of Observational evidence for on-shelf transport of Warm Deep Water driven by dense water export in the Weddell Sea by Darelus et al.

This study uses observations from eastern Weddell Sea to investigate onshore (i.e. across the Antarctic shelf break) transport of Warm Deep Water (WDW), the regional version of Circumpolar Deep Water (CDW). The work is motivated by a recent modelling study that finds a close regional link between Dense Shelf Water (DSW) export and CDW inflow. The finding from the modelling study changes profoundly our view on cross-shelf heat transport and where it occurs. The present study provides the first observational evidence for the mechanism presented in the previous model study. The conclusions are based on

- (i) temperature observations which reveal a shallower upper limit of the WDW layer in the vicinity of the DSW outflow compared to the continental shelf break further east where no DSW export exists, and
- (ii) the onshore direction of the velocity vector in the WDW layer with the largest velocities just east (upstream) of the DSW outflow, matching the expectations from the modelling work.

The manuscript is concise and well written, the figures are appropriate, and the data and methodology are described in detail. While the data interpretation and conclusion are robust, my main comment is on the missing temporal correlation between the DSW outflow and WDW inflow, which I detail below, followed by specific and technical comments.

We thank reviewer two for their comments – and answer each of them below. When the shelf waves were filtered out, as suggested by both reviewers, the records correlated, in accordance with the mechanism proposed by Morrison et al.

Main comment

L114-118: The missing temporal correlation between the downslope and upslope flow is surprising as it is an important argument in Morrison et al. (2020). The conclusion of the study would be more robust if the mismatch in the temporal variability could be resolved. In the model, the temporal correlation is largest on the subdaily timescale, do the moorings capture this timescale? Can you discuss why there is no temporal correlation in the observations? Did you try filtering the data to remove the dominating imprint of the continental shelf waves and then calculate the correlation?

We were admittedly blinded by the variability linked to the shelf waves and did not think about filtering them out. Morrison et al. show a significant correlation for their 3h-resolution records of transport – but their model does not include tides, and they are in a region where the bottom slope is too steep for the outflow to generate shelf waves (Han et al., 2022). Following your comments - and those of the first reviewer – we now present low-pass filtered currents from the two concurrent moorings (New Fig. 2) and show that there is significant coherence at periods longer than ten days and that the low-pass filtered currents are negatively correlated ($r=-0.67$, $p<0.01$ at three days lag). When there is higher than normal downslope flow close to the ridge, there is an anomalously high upslope current at the mooring 25 km east of the ridge – consistent with the mechanism outlined in the Morrison et al. paper. Visual inspection of their low pass filtered transports (black lines in their Fig 4) suggest that their transports are anti-correlated also at longer time scales. (They do not give the correlation for the low-pass filtered currents).

The mooring records have hourly resolution – but at sub-daily time scales tidal currents dominate the records.

Do the continental shelf waves promote any onshore WDW transport in the Weddell Sea?

This is a very interesting question deserving its own dedicated analysis. Now that we clarify the correlation between the low-pass filtered currents and that this frequency band is distinct from that of the continental shelf waves, we prefer to leave the potential impact of shelf waves for future studies.

Specific comments

L057-058: On the statement that the Weddell and Ross Gyres move CDW closer to the continental shelf. CDW can also be found elsewhere on the continental shelf (e.g. Amundsen and Bellingshausen Seas, or near the Totten Ice Shelf in East Antarctica). How does this fit to the statement?

We agree and have expanded the text:

The oceanic heat originates from the Circumpolar Deep Water (CDW) that circumnavigates the continent in the Antarctic Circumpolar Current (ACC). The warm waters approach the continental shelf of the marginal seas either due to the proximity of the ACC, e.g. in the Amundsen Sea, or as part of subpolar gyres, e.g. the Weddell and Ross gyres. In the Weddell Sea, which is the focus here, the CDW mixes with relatively fresh waters and cools to form Warm Deep Water (WDW). WDW is transported along the continental slope of the southern Weddell Sea in the Antarctic Slope Current, and it is separated from the continental shelf by the Antarctic Slope Front (Thompson et al, 2018).

L063-064: Please add reference for dense water formation.

We've added a reference to Orsi et al. (1999).

L075-076: (i) Do observations of other onshore CDW transport processes exist? Adding the information here might help with my comment on L159-160.

We now mention that onshore CDW transport is typically associated with troughs on the warm shelves and eddy transports. The new text reads:

On-shelf heat fluxes are typically associated with troughs on the warm shelves (e.g. Walker et al, 2007; Arneborg et al, 2012) and eddy-transports (e.g. Stewart and Thompson, 2015)

(ii) Morrison et al. (2020) discuss that Orsi et al. (2009) and Williams et al. (2010) report concurrent CDW inflow and DSW outflow. Can you comment on those observations and acknowledge them if you assess them as insightful?

Indeed, these references support the mechanisms suggested by Morrison et al, in that the CTD-observations described show CDW-intrusions that are co-located with dense outflows. We've now included them in the ms, as suggested:

While there are examples of CDW intrusions being co-located with dense shelf water export (e.g. Orsi and Wiederwohl, 2009; Williams et al., 2010), the process for on-shelf transport suggested by Morrison et al. (2020) is hitherto unobserved.

L103-105: Can you comment on the fact that most of the flow captured by the CTD section is in along-slope direction (and not downslope)? Does this matter?

We presume that you refer to the CTD section shown in Supplementary Fig 1. The CTD-section in Supplementary Fig 1 was obtained roughly perpendicular to the ridge and, on a larger scale, parallel to the continental slope. The current associated with the sloping isotherms/isopycnals in Supplementary Fig. 1 is the downslope (parallel to the ridge) flow observed at D1 (Fig 1d).

We now include a map in Supplementary Fig. 1 to clarify this.

L108-109: The moorings show along-slope flow at depth and onshore flow in the WDW layer. Can you comment on the co-location of the flow? The mechanism described in the text so far suggests a spatial offset between downslope ISW and upslope WDW transport.

There is a spatial offset between the downslope flow of ISW and the upslope flow of WDW – we observe downslope flow at D1 and upslope flow at moorings east of the ridge at D2 (black triangle and black square in Fig 1d). The two moorings, D1 and D2, are separated by roughly 25 km, in close agreement with the 20 km offset found in the simulations by Morrison et al. (2020) (as noted by the reviewer below). For the other moorings upstream of the ridge (black dots in Fig 1D), we also observe the upslope front above the plume, though weaker. At these locations, the velocity in the plume is alongslope, as expected from a dense gravity current under the influence of rotation. According to Morrison et al. (2020) the upslope flow should be barotropic, so that there must be a baroclinic component associated to the dense plume that is larger than the barotropic forcing that causes the upslope flow in the upper layer. Note that we use a different scale for the flow above/outside the plume, which at most moorings is much weaker than the alongslope flow within the plume. We clarify this in the ms (L 135 and on).

L111-112: Do you expect to find the maximum upslope velocity to be at this location? Or is it impossible to make a statement on this due to lack of observations in the area? The 25 km is close to the 20 km that Morrison et al. (2020) find.

The available moorings do not allow us to determine where the maximum of the upslope current is located, but we now mention the scale (20km) given by Morrison et al. in the ms:

The largest upslope current (0.05 m s^{-1}) is observed at a mooring located roughly 25 km east of the ridge at 1800 m depth (D2, Fig 1d), in close agreement with the 20 km offset found in the simulations by Morrison et al (2020).

L127: Suggest adding 34°W longitude to Fig. 1 to help with orientation.
 34W is now marked in Fig 1

L148-151: Please add a statement on the missing temporal correlation between the downslope and upslope flow.

See the answer above and the new Fig. 2. We now show temporal correlation at timescales longer than ten days.

L159-160: Where does the remainder (90%) of the WDW-flux occur? At the eastern part of the FT as described in Darelius et al. (2004) and Ryan et al. (2017)?

We realized when re-reading that the 3 Sv transport in Nicholls et al refers to mWDW with a mean temperature of only -1.76C. Mixing WW at -1.85C with WDW (at 0.5C), you only need about 5% of WDW – so 0.3SV WDW supplied by the ridge would be sufficient. The text in the ms has been changed accordingly:

Assuming that the associated on-slope flow of WDW is of the same order, as suggested by Fig. 3 in Morrison et al (2020), it amounts to about 4% of the total circumpolar CDW flux in the simulations by Morrison et al (2020) and all (see methods) of the WDW flux required to feed the local dense water production (Nicholls et al, 2009).

Details are given in methods:

Nicholls et al (2009) stated that an onshelf transport of modified WDW (mWDW, potential temperature of $\theta = -1.76^{\circ}\text{C}$, and practical salinity of $S = 34.5$) of about 3 Sv is required to sustain the dense water production on the continental shelf. If the mWDW consists of a mixture of WDW ($\theta = 0.5^{\circ}\text{C}$) and Winter Water (WW, -1.85°C), then less than 5%, or 0.15 Sv, of the mWDW would be WDW.

The warm inflow described by (Darelius et al., 2016; Ryan et al., 2017) occurs east of the Filchner Trough and does not directly contribute to DSW production, which occurs on the continental shelf east of the trough (Nicholls et al, 2009). Observations suggest that the southward flow of mWDW that feeds the DSW production on the Ronne shelf occurs within the central trough on the Ronne side of the continental shelf (Nicholls et al., 2008, 2009).

Morrison et al. (2020) suggest the majority of the onshore CDW transport is connected to the DSW export, do the observations presented here support the model findings?

We cannot estimate volume transports nor compare the relative contribution by different processes / in different regions. The presented observations support the mechanism proposed by Morrison et al, but cannot, on their own, support the circumpolar findings from the model.

L180-181: Do you expect the summer bias of the CTD observations to affect the findings of this study?

No, there is a seasonality in the properties of the dense outflow across the sill of the Filchner Trough, but no observed seasonality in its velocity (Darelius et al., 2014).

L340: Is the slope front current different to the ASC? If not, please use the same name throughout the manuscript.

We now use the Antarctic Slope Current consistently throughout the ms.

Figure 1

- Colormap for Fig 1a-b): Suggest explaining in figure caption that -0.5°C represents the threshold between ISW and WDW and that is why a divergent colormap was chosen which is centred around -0.5°C .

This is now mentioned in the caption: The divergent colormap highlights the depth of the thermocline, with Winter Water above (in blue) and (modified) WDW below (in red)

- Fig 1d): Grey temperature section in Fig. 1d) is difficult to see, would a different colour (green?) improve the visibility?

The location of the section is now shown in green, as suggested.

- L212: The location of CTD-stations on the continental slope is marked. Are they marked by the white dots? Please specify and mention what the white dots represent otherwise.

The location of CTD-stations on the continental slope is marked with white dots – this is now stated in the caption.

- Suggest adding explanation of mooring locations (white circle and square) in Fig. 1e) to figure caption.

We've now used the mooring names (D1 & D2) in the text and added a legend to Fig. 1e.

Figure 2

- What is the shading in Fig 1a) indicating? Please describe in figure caption.

The grey shading indicates the longitudes of the Filchner Trough. This is now stated in the caption

- What are the black dots in Fig. 1f)?

We presume you refer to (the old) Fig. 2f. The black dots indicate the position of the profiles shown in Fig2e, i.e., the profiles from May. This is stated in the caption: Map showing the position of the profiles in panels d (red/blue dots) and e (black dots).

Figure S1

- Should the minimum SSH not be centred around the dense water plume? (Assuming conservative temperature serves as a tracer for the dense plume.)

Conservative temperature is a tracer for the dense plume – this is now stated in the caption. We have adjusted the SSH signature in (a) as suggested.

- The section shown is oriented across the continental slope and therefore approximately orthogonal to the shelf break. The surface pressure argument brought forward in Morrison et al. (2020) and described in L077-085 assumes a SSH change across a trough/ridge (parallel to the shelf break). Why is the SSH shown schematically in Fig. S1a) relevant?

The section is not oriented across the continental slope but across the ridge; hence, on a larger scale, the CTD section is parallel to the slope. Darelius & Wählin (2007) explain that ridges can – just like the canyons discussed in Morrison et al (2020)- steer dense water downslope. The dense water “leaning” on the ridge in Fig S1 flows down the slope – this is the current observed at mooring D1. Figure S1 is hence the ridge analogue to Fig 5a in Morrison et al (2020).

To clarify this in the paper, we have a) included a map showing the location and orientation of the CTD section and b) inserted the word “Ridge” in Fig. S1b.

Technical comments

L069-070: Check reference format

Corrected

L097: It should be Fig. 1e)

Corrected

L110: Add info on panel: Fig. 1e). Suggest adding information that upslope flow is shown in red.
The caption states that currents observed above / outside the plume are shown in red.

L112: It should be Fig. 1d)

Corrected

L119: Add info panel: Fig. 1b-c)

Corrected

L121: Correct to: extends

Corrected

L122: Correct to: vicinity.

Corrected

L132: Delete with east of the

Corrected

L136: Remove Front

Corrected

L152: Rewrite to: on-shelf flow

Corrected

L159: Does the total flux refer to the total circumpolar flux?

Yes, this is now specified in the ms

L302: Remove capital letter in we

Corrected

L320-321: Rewrite to: Antarctic Slope Current (capital letters) and remove ASC acronym unless it is used in the manuscript.

We now use "Antarctic Slope Current" throughout the ms.

Figure 1

L210: Replace slope front with slope current or Antarctic Slope Current (see comment on L340)

Corrected

Please change the brackets of the axis labels to square brackets for consistency with other figures.

Figure 2

L261: Specify panel where black dots are shown: Fig. 1d)

The location of the profiles are now shown in Fig 2f and this is stated in the caption

L 264: Rewrite to: Temperature profiles from d) March and e) May

Corrected

L266: Rewrite to: f) Map showing

Corrected

Figure S1

Please use larger font size.

Please change the brackets of the axis labels to square brackets for consistency with other figures.
Corrected

Figure S2

I suggest being consistent with the orientation of the vertical depth coordinate, i.e. change the depth values to positive as in Fig 1-2.

Corrected

Table 1

L323: Consider highlighting T(V) ins. height in table caption in italic or asterisks to improve readability.

Corrected

Mentioned (new) references

Orsi, A. H., Wiederwohl, C. L. (2009), A recount of Ross Sea waters. *Deep-Sea Res. II* 56, 778–795, doi: 10.1016/j.dsr2.2008.10.033.

Williams, G. D., S. Aoki, S. S. Jacobs, S. R. Rintoul, T. Tamura, and N. L. Bindoff (2010), Antarctic Bottom Water from the Adélie and George V Land coast, East Antarctica (140–149°E), *J. Geophys. Res.*, 115, C04027, doi:10.1029/2009JC005812.

References used in the answers

Darelius, E., Fer, I., & Nicholls, K. W. (2016). Observed vulnerability of Filchner-Ronne Ice Shelf to wind-driven inflow of warm deep water. *Nature Communications*, 7:12300. <https://doi.org/10.1038/ncomms12300>

Darelius, E., Strand, K. O., Østerhus, S., Gammelsrød, T., Årthun, M., & Fer, I. (2014). On the seasonal signal of the Filchner Overflow, Weddell Sea, Antarctica. *Journal of Physical Oceanography*, 44, 1230–1243. <https://doi.org/10.1175/JPO-D-13-0180.1>

Han, X., Stewart, A. L., Chen, D., Lian, T., Liu, X., & Xie, X. (2022). Topographic Rossby Wave-Modulated Oscillations of Dense Overflows. *Journal of Geophysical Research: Oceans*, 127(9). <https://doi.org/10.1029/2022jc018702>

Nicholls, K. W., Boehme, L., Biuw, M., & Fedak, M. A. (2008). Wintertime ocean conditions over the southern Weddell Sea continental shelf, Antarctica. *Geophysical Research Letters*, 35. <https://doi.org/10.1029/2008GL035742>

Nicholls, K. W., Østerhus, S., Makinson, K., Gammelsrød, T., & Fahrbach, E. (2009). Ice-Ocean Processes over the Continental Shelf of the Southern Weddell Sea, Antarctica: A Review. *Reviews of Geophysics*, 47, 1–23. <https://doi.org/10.1029/2007RG000250>

Orsi, A. H. H., Johnson, G. C. C., & Bullister, J. L. L. (1999). Circulation, mixing, and production of Antarctic Bottom Water. *Progress in Oceanography*, 43(1), 55–109. [https://doi.org/10.1016/S0079-6611\(99\)00004-X](https://doi.org/10.1016/S0079-6611(99)00004-X)

Ryan, S., Hattermann, T., Darelius, E., & Schröder, M. (2017). Seasonal Cycle of Hydrography on the Eastern Shelf of the Filchner Trough, Weddell Sea, Antarctica. *Journal Geophysical Research - Oceans*, 122. <https://doi.org/10.1002/2017JC012916>

Thompson, A. F., Stewart, A. L., Spence, P., & Heywood, K. J. (2018). The Antarctic Slope Current in a Changing Climate. *Reviews of Geophysics*. <https://doi.org/10.1029/2018RG000624>

REVIEWER COMMENTS

Reviewer #1 (Remarks to the Author):

Following the latest round of revisions, I believe the authors have addressed all of my comments.

Reviewer #2 (Remarks to the Author):

Review of revised manuscript Observational evidence for on-shelf heat transport driven by dense water export in the Weddell Sea by Darelius et al.

I thank the authors for incorporating the previous feedback. In particular, the revised analysis of the temporal correlation between the downslope and upslope flow now supports the modelling findings and improves the conclusions of the manuscript. I do have a few comments, which I detail below, mostly regarding the new analysis of the temporal correlation. I recommend the manuscript to be accepted for publication after these points have been addressed.

Specific comments

L162: The text says that there is a significant coherence on longer time scales (> 10 days). I assume this statement is based on Fig 2a) and refers to the values above the dashed line (indicating the significance) on the left side of the graph. I do not understand why 10 days is picked as the threshold, it should be 15 or 16 days. The coherence is NOT significant between 10-15 days. (Note that I think it is okay to use a 10-day threshold for the low-pass filtering in Fig 2b.)

L162: On the significant coherence between the D1 and D2 raw time series: Please mention somewhere that there are also peaks at shorter time scales from the diurnal and semi-diurnal tide. I understand that the significance on this time scale does not matter for the mechanism described in the manuscript, but it is confusing that the information is omitted.

L162: Fig 2a) is never referred to in the text, please incorporate e.g. here.

L164: On the time lag between D1 and D2: Which time series is leading? It looks like D2 is leading by 3 days. Please discuss why the onshore flow is leading the offshore flow. Is the lag just an artifact of the exact locations of the two moorings with D2 being closer to the shelf break compared to D1?

Fig 2: Please explain the meaning of "mab" (meters above bottom?) in caption.

Technical comments

L164: Move "Fig 2b" into the bracket.

L242: Remove "We note that".

Fig S1 caption: Replace "articl" with "article".

Dear reviewer,

We thank you again for your thorough review, which has further improved the manuscript. We have addressed your comments, and answer them one by one below (in italics).

Best regards,
Elin Darelius and co-authors

Reviewer #1 (Remarks to the Author):

Following the latest round of revisions, I believe the authors have addressed all of my comments.

Reviewer #2 (Remarks to the Author):

Review of revised manuscript *Observational evidence for on-shelf heat transport driven by dense water export in the Weddell Sea* by Darelius et al.

I thank the authors for incorporating the previous feedback. In particular, the revised analysis of the temporal correlation between the downslope and upslope flow now supports the modelling findings and improves the conclusions of the manuscript. I do have a few comments, which I detail below, mostly regarding the new analysis of the temporal correlation. I recommend the manuscript to be accepted for publication after these points have been addressed.

Specific comments

L162: The text says that there is a significant coherence on longer time scales (> 10 days). I assume this statement is based on Fig 2a) and refers to the values above the dashed line (indicating the significance) on the left side of the graph. I do not understand why 10 days is picked as the threshold, it should be 15 or 16 days. The coherence is NOT significant between 10-15 days. (Note that I think it is okay to use a 10-day threshold for the low-pass filtering in Fig 2b.)

You are right, and we have change to 15 days (but kept the low-pass filtering at ten days, as suggested). Note that we, by mistake, had calculated correlation using a 12-day threshold for the low-pass filter. We now consistently use a 10-day filter and the numbers given for correlation, significance etc. have therefor changed slightly, but it does not affect the conclusions in any way. For example, the correlation between the filtered time series from D1 and D2 changes from $r=-0.67$ to $r= -0.65$.

L162: On the significant coherence between the D1 and D2 raw time series: Please mention somewhere that there are also peaks at shorter time scales from the diurnal and semidiurnal tide. In understand that the significance on this time scale does not matter for the mechanism described in the manuscript, but it is confusing that the information is omitted. *We agree, and have changed the text to: Still, they show significant coherence on tidal frequencies and, more relevant here, on time scales longer than about 15 days.*

L162: Fig 2a) is never referred to in the text, please incorporate e.g. here.

We now refer to Fig 2a in the text, as suggested

L164: On the time lag between D1 and D2: Which time series is leading? It looks like D2 is leading by 3 days.

We now specify that D2 is leading: ...with the highest correlation found for a lag of three days with D2 leading

Please discuss why the onshore flow is leading the offshore flow. Is the lag just an artifact of the exact locations of the two moorings with D2 being closer to the shelf break compared to D1?

We cannot explain the lag, which seems to be relatively consistent throughout the time series as a positive lag is observed for all eight 60-days long, 50% overlapping subsections of the record (Shown in the new Supplementary Figure 2). We hypothesize (in the first section of the discussion) that the lag is linked to the fact that we are correlating velocities from point measurements at different isobaths while Morrison et al (2020) compare up and downslope transports.

Fig 2: Please explain the meaning of “mab” (meters above bottom?) in caption.

We now write “meters above bottom” in the caption

Technical comments

L164: Move “Fig 2b” into the bracket.

Corrected

L242: Remove “We note that”.

Corrected

Fig S1 caption: Replace “articl” with “article”.

Corrected